# Deep Supramolecular Language Processing for Co-crystal Prediction

Rebecca Birolo, [1 2]  Rıza Özçelik, [1 3]  Andrea Aramini, [4]  Roberto Gobetto, [2]  Michele R. Chierotti, [2]
Francesca Grisoni* [1 3]

## Abstract

Approximately 40% of marketed drugs exhibit suboptimal pharmacokinetic profiles. Co-crystallization, where pairs of molecules form a multicomponent crystal, constitutes a promising strategy to enhance physicochemical properties without compromising the pharmacological activity. However, finding promising co-crystal pairs is resource-intensive, due to the vast number of possible combinations. We present DeepCocrystal, a novel deep learning approach designed to predict co-crystal formation by processing the 'chemical language' from a supramolecular vantage point. Rigorous validation of DeepCocrystal showed a balanced accuracy of 78% in realistic scenarios, outperforming existing models. By leveraging properties of molecular string representations, DeepCocrystal can also estimate the uncertainty of its predictions. We harness this capability in a challenging prospective study, and successfully discovered two novel co-crystal of diflunisal, an anti-inflammatory drug. This study underscores the potential of deep learning – and in particular of chemical language processing – to accelerate co-crystallization, and ultimately drug development, in both academic and industrial contexts.

## 1. Introduction

Co-crystallization enables the optimization of the pharmacokinetic properties of active pharmaceutical ingredients (APIs) (Duggirala et al., 2016; Thayyil et al., 2020). Via

co-crystalization, supramolecular interactions between the API and another molecule (coformer) are established to form a multicomponent crystal (Desiraju, 1995) (Fig. 1a). The resulting co-crystal preserves the bioactivity of the lead molecule while enhancing desirable properties, such as solubility, and stability. Owed to the high number of possible combinations, finding the optimal coformer for a given API is far from trivial, and ultimately relies on a labor- and time-intensive process based on trial and error (Ngilirabanga & Samsodien, 2021; Cappuccino et al., 2022).

Machine learning – which extracts relevant information from chemical datasets(Artrith et al., 2021) – can aid in prioritizing API-coformer pairs for co-crystallization(Sarkar et al., 2020; Molajafari et al., 2024; Wang et al., 2020; Yang et al., 2023; Kang et al., 2023). Current methods, however, might struggle to generalize to previously unseen molecules(von Essen & Luedeker, 2023). This is in part due to limitations of training datasets, which are unrealistically imbalanced towards existing co-crystals (Heng et al., 2021). Therefore, there is a need for approaches that are more robust to data imbalance and demonstrate stronger generalizability to previously unseen molecules.

Here we introduce DeepCocrystal, a novel deep learning approach designed to learn the "supramolecular language" of co-crystallization. Supramolecular chemistry can be viewed as a language(Cragg & Cragg, 2010; Lehn, 1988; Brock & Dunitz, 1994): atoms ('letters') form molecules ('words'), whose combinations give rise to supramolecular interactions ('sentences'). Building on this analogy, we extend current chemical language processing techniques(Hirohara et al., 2018; Kimber et al., 2018; van Tilborg et al., 2022; Öztürk et al., 2020) — which predict molecular properties from single string representations(Weininger, 1988; Krenn et al., 2022) — to predicting supramolecular interactions between pairs of molecules (i.e., co-crystallization).

DeepCocrystal represents single molecules (API and coformer) as SMILES (Simplified Molecular Input Line Entry Systems(Weininger, 1988)) strings (Fig. 1b), whose chemical information is combined to predict whether they form co-crystals. Thanks to intriguing properties of the SMILES language(Bjerrum, 2017), DeepCocrystal addresses the data imbalance and estimates prediction uncertainty, pivotal for

[1]Institute for Complex Molecular Systems, Department of Biomedical Engineering, Eindhoven University of Technology, Eindhoven, The Netherlands. [2]Department of Chemistry and NIS Centre, University of Torino, Torino, Italy. [3]Centre for Living Technologies, Alliance TU/e, WUR, UU, UMC Utrecht, Utrecht, The Netherlands. [4]Research and Early Development, Dompé Farmaceutici S.p.A, L'Aquila, Italy. Correspondence to: F. Grisoni <f.grisoni@tue.nl>.

*Accepted at the 1st Machine Learning for Life and Material Sciences Workshop at ICML 2024.* Copyright 2024 by the author(s).

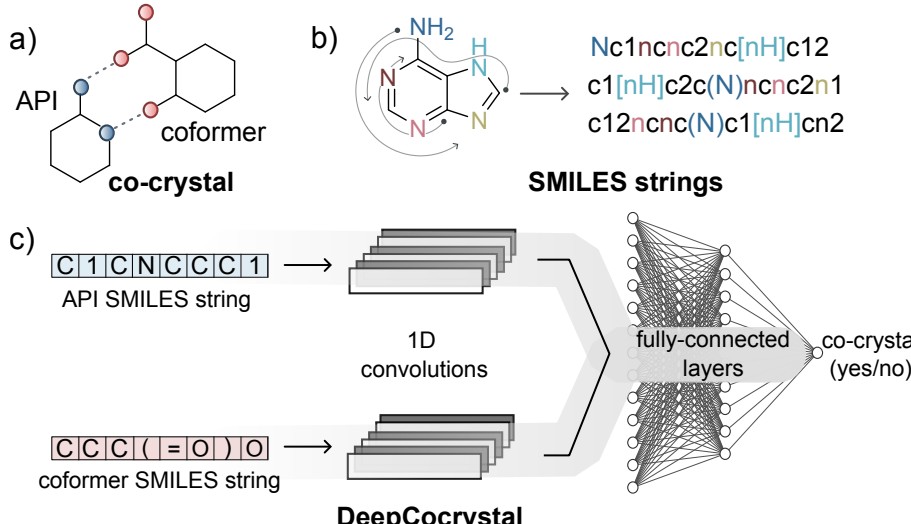

*Figure 1. Overview of key elements of DeepCocrystal for co-crystal prediction.* **a)** The co-crystallization between an active pharmaceutical ingredient (API) and a coformer involves the formation of a multicomponent crystalline structure (co-crystal), in which the API and coformer are held together by non-covalent interactions. **b)** SMILES strings, which convert a molecular graph into one string. One molecule can be represented by many different SMILES strings, based on the starting (non-hydrogen) atom and the chosen direction for graph traversal. **c)** DeepCocrystal represents API and coformers via SMILES strings and passes them through 1-dimensional (1D) convolutions. Fully-connected layers are then used to predict the co-crystallization output as a continuous number between 0 and 1, which can be then discretized (with a cut-off of 0.5) to perform a prediction ("negative" pair if below, and "positive" pair otherwise).

prospective applications.

In this work, DeepCocrystal shows superior performance and generalization capacity than existing approaches(Mswahili et al., 2021; Devogelaer et al., 2020; Liang et al., 2024; Jiang et al., 2021). When applied prospectively to identify coformer candidates, all high-certainty predictions of DeepCocrystals were confirmed experimentally – leading to the identification of two previously unreported diflunisal co-crystals. To the best of our knowledge, this is the first application of "supramolecular language" processing to predict co-crystallization – opening novel opportunities in supramolecular chemistry.

## 2. Results and Discussion

### 2.1. DeepCocrystal architecture

DeepCocrystal has at its core Convolutional Neural Networks (CNNs) (LeCun et al., 1998) for 'chemical language' processing. CNNs are a class of deep learning models commonly used for processing sequences of text (Yin et al., 2017). Via convolution – which involves sliding a filter (kernel) over the input text – CNNs can capture information and features at different levels of abstraction, and progressively aggregate it to provide a prediction. DeepCocrystal leverages SMILES(Weininger, 1988) strings as an input, which are derived from traversing a molecular graph from a non-hydrogen atom, and annotating atoms and bonds with

specific symbols (Fig. 1b). CNNs have been previously applied to predict the properties of single molecules from their SMILES strings(Hirohara et al., 2018; Kimber et al., 2018; van Tilborg et al., 2022).

DeepCocrystal extends traditional chemical language processing approaches beyond the 'one-molecule-one-property' paradigm, to learn simultaneously from the SMILES strings of *pairs* of molecules (*i.e.*, API-coformer pairs). In particular, DeepCocrystal uses two separate CNNs to learn 'latent representations' of the input molecular structures (of each API and coformer), and then aggregates this information via a fully-connected neural network, to predict the potential co-crystallization of the input pair (Fig. 1c). Via the DeepCocrystal architecture, the co-crystalization potential of any molecular pair is predicted as a number between 0 (negative) and 1 (positive).

In this work, every API-coformer pair was presented to the network twice, once per every separate CNN, as previously suggested (Jiang et al., 2021; Kang et al., 2023). This strategy allows artificially increasing the number of inputs available for model training. Moreover, we experimented with different SMILES string variations, to serve as input for DeepCocrystal. In particular, we experimented with (a) *canonical SMILES*, which provide a univocal string per every molecular structure via standardization algorithm(Schneider et al., 2015), and (b) *'randomized' SMILES*, which can provide a different SMILES string

based on the chosen starting atom and the graph traversal route (Fig. 1b). Randomized SMILES strings were used to perform 'data augmentation' (Bjerrum, 2017), *i.e.*, to artificially inflate the number of data available for training by using multiple SMILES for a single molecule.

## 2.2. DeepCocrystal training and validation

To train and validate DeepCocrystal, we collected and manually curated a dataset of experimentally-determined co-crystal structures, from (a) the Cambridge Structural Database (Groom et al., 2016) and (b) existing co-crystal literature (Shen, 1983a; Aakeröy et al., 2011; Grecu et al., 2014b;a; Roca-Paixão et al., 2019; Jiang et al., 2021). Moreover, a set of in-house experiments was conducted to measure the co-crystalization of additional molecular pairs. The collected dataset comprises a total of 6632 API-coformer pairs, of which 5240 (79%) are co-crystals ("positive") and 1392 (21%) are physical mixtures ("negative", *i.e.,* no observed co-crystallization).

The training, validation and internal test sets were created by stratified splits of this dataset (10 randomly sampled subsets with 10% molecules in validation and test folds). In addition to using canonical SMILES as input, we also experimented with different levels of augmentation: (a) [positive:negative = 1:4], where one randomized SMILES string is used for every molecule in a "positive" pair, and four SMILES are used for molecules in "negative" pairs, and (b) [positive:negative = 2:7], where a two-fold and a seven-fold augmentation are used for the SMILES strings of positive and negative pairs, respectively. Each model variant was evaluated for its classification performance(Ballabio et al., 2018) (Table 1), *i.e.*, via Recall (ability to correctly classify positive pairs), Specificity (ability to correctly classify negative pairs) and Balanced Accuracy (overall performance). These metrics were computed by considering predictions lower than 0.5 as a "negative", or "positive" otherwise.

All DeepCocrystal variants reached a Balanced Accuracy above 88%, with the 2:7 augmentation performing the best. When looking at class performance, different trends can be observed. In identifying "positive" pairs, canonical SMILES lead to the best performance (up to 5% increase in recall). All DeepCocrystal variants have a good capacity to recognize "positive" pairs, with 1:4 and 2:7 augmentations showing comparable performance. DeepCocrystal trained on canonical SMILES showed a significantly higher Recall than the two augmented models (Wilcoxon signed-rank test, $p < 0.05$). On the contrary, the 2:7 SMILES augmentation significantly improves the ability to identify negative pairs (Wilcoxon signed-rank test, $p < 0.05$), resulting in an 8% increase in specificity compared to the canonical version. This evidence highlights how SMILES augmentation on the negative class, can aid in mitigating the data unbalance.

## 2.3. Model benchmarking

The predictive performance of DeepCocrystal was then evaluated on an external test set, which was manually curated by combining public data with in-house experimental co-crystallization results of selected APIs (*see* Materials and Methods). This external set contained 364 pairs (129 are co-crystals and 235 non-co-crystals), with a lower substructure similarity(Rogers & Hahn, 2010) to the training set than the internal test set – constituting a more challenging validation set.

DeepCocrystal was benchmarked with four existing approaches: (i) CCGNet(Jiang et al., 2021), which relies on graph neural networks to perform a prediction; (ii) CC-Descriptor ML, which relies on an array of 'classical' machine learning models trained on co-crystal descriptors (Liang et al., 2024); (iii) Descriptor-DNN, based on a fully-connected neural network trained on molecular descriptors (Mswahili et al., 2021); and (iv) Fingerprint-DNN, a fully-connected neural network trained on extended connectivity fingerprints (Devogelaer et al., 2020; Chen et al., 2024). To ensure comparability and account for the lack of provided code, data, and/or hyperparameters, we re-implemented and trained Descriptor-DNN and Fingerprint-DNN, using the same dataset as DeepCocrystal (*see* Materials and Methods).

DeepCocrystal consistently outperformed the benchmarks (Table 1). DeepCocrystal, in its augmented 2:7 configuration, achieved 15%-21% higher balanced accuracy and 12%-56% higher specificity than the benchmarks, albeit with a moderate recall reduction (of up to 15% lower). These results indicate that DeepCocrystal finds a better trade-off between positive and negative prediction power than the benchmarks, which are unbalanced toward the positives. Furthermore, the SMILES augmentation increased the balanced accuracy by 10% and 19%, respectively for 1:4 and 2:7 augmentation levels, compared to using canonical SMILES strings, indicating a higher generalization potential provided by learning from different SMILES versions of the same molecule.

## 2.4. Uncertainty estimation

To extend the applicability of DeepCocrystal to real-world scenarios, we equipped it with an estimate of its (un)certainty. We represented each molecular pair with ten different (pairs of) SMILES strings, and used DeepCocrystal (2:7) predictions to estimate uncertainty. Considering the predictions on SMILES ensembles (*i.e.*, by average prediction, Fig. 2), allows detecting some of the model errors.

We tested two ways of estimating the DeepCocrystal's uncertainty starting from its predictions on the 'molecular-pair ensemble' (*i.e.*, 10-fold SMILES repetitions for each molecular pair): (a) *Majority voting*, whereby the number of

*Table 1. Performance of DeepCocrystal.* DeepCocrystal was tested on two test sets, one internal and one external. The internal test sets was composed of 664 molecular pairs, which were sampled by stratified splits of the collected dataset. The external set was composed of 364 pairs collected in a second phase of the project, and containing more structurally diverse molecular pairs. The external test set was used to benchmark DeepCocrystal with existing literature models (*i.e.*, Fingerprint-DNN, Descriptor-DNN, CC-Descriptor-ML, and CCGNet(Mswahili et al., 2021; Devogelaer et al., 2020; Liang et al., 2024; Jiang et al., 2021)). Balanced accuracy (global performance), recall (performance on "positive" pairs), and specificity (performance on "negative" pairs) are reported for each set and each model (the closer to 100%, the better). The best performance per metric is highlighted in boldface for each considered test set.

| Test set | Model | BAcc | Recall | Specificity |
|---|---|---|---|---|
| Internal | DeepCocrystal - canonical | $88\% \pm 2\%$ | $\mathbf{96\% \pm 1\%}$ | $79\% \pm 6\%$ |
| | DeepCocrystal - augmented (1:4) | $88\% \pm 2\%$ | $91\% \pm 2\%$ | $86\% \pm 3\%$ |
| | DeepCocrystal - augmented (2:7) | $\mathbf{89\% \pm 2\%}$ | $92\% \pm 2\%$ | $\mathbf{87\% \pm 3\%}$ |
| External | DeepCocrystal - canonical | 59% | **93%** | 26% |
| | DeepCocrystal - augmented (1:4) | 69% | 71% | 66% |
| | DeepCocrystal - augmented (2:7) | **78%** | 75% | **81%** |
| | CCGNet (Jiang et al., 2021) | 60% | 51% | 69% |
| | CC-Descriptor-ML[a] (Liang et al., 2024) | 63% | 79% | 48% |
| | Descriptor-DNN (Mswahili et al., 2021) | 63% | 84% | 41% |
| | Fingerprint-DNN (Devogelaer et al., 2020) | 57% | 90% | 25% |

[a]Performance computed by excluding five molecular pairs that were used for model training.

*Table 2. Uncertainty estimation with DeepCocrystal.* External test set molecules were represented as 10 SMILES strings each before prediction (using DeepCocrystal 2:7). Two approaches were considered to estimate uncertainty, *i.e.*, majority voting, which picks the most frequent class among the predictions (per molecular pair), and standard deviation computed on the individual model predictions per each pair. Different uncertainty thresholds on each approach were analyzed for their effect on the model performance, as well as on the number of molecular pairs predicted. The number and percentage of predicted pairs (*i.e.*, predictions below the considered thresholds), balanced accuracy (BAcc), recall, and specificity are reported. DeepCocrystal on canonical SMILES (which is invariant to augmentation and cannot be used for uncertainty estimation) was used as a performance baseline. The best performing models per metric are highlighted in boldface.

| SMILES input | Method | Thr. | No. Pairs (%) | BAcc | Recall | Specificity |
|---|---|---|---|---|---|---|
| Canonical | - | - | 364 (100%) | 78% | 75% | 81% |
| Augmented (10-fold) | Major. | $\geq 50\%$ | 364 (100%) | 76% | 75% | 77% |
| | Major. | $\geq 60\%$ | 348 (96%) | 77% | 75% | 79% |
| | Major. | $\geq 70\%$ | 313 (86%) | 79% | 77% | 82% |
| | Major. | $\geq 80\%$ | 287 (79%) | 82% | 79% | 84% |
| | Major. | $\geq 90\%$ | 254 (70%) | 84% | 82% | 86% |
| | Major. | $= 100\%$ | 218 (60%) | 87% | **86%** | 89% |
| | St. dev. | $\leq 0.50$ | 364 (100%) | 76% | 75% | 77% |
| | St. dev. | $\leq 0.40$ | 351 (96%) | 77% | 76% | 78% |
| | St. dev. | $\leq 0.30$ | 275 (76%) | 82% | 80% | 83% |
| | St. dev. | $\leq 0.20$ | 227 (62%) | 86% | 85% | 87% |
| | St. dev. | $\leq 0.10$ | 191 (52%) | **88%** | **86%** | 90% |
| | St. dev. | $\leq 0.05$ | 161 (44%) | **88%** | 84% | **91%** |

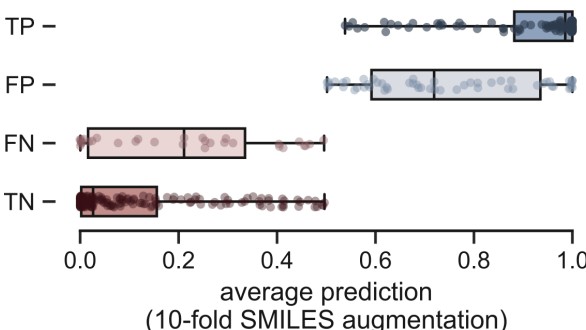

*Figure 2. Relationship between DeepCocrystal predictions and classification performance.* The SMILES of external test set samples were augmented 10 times and the average prediction was computed per API-coformer pair. Such average prediction was used to classify the molecular pairs based on a cut-off of 0.5 (negative if below, and positive otherwise). Molecular pairs were by comparing their true class with the predicted class: TP = True Positive; FP = False Positive; FN = False Negative; TN = True Negative. Box plots depict the distribution of DeepCocrystal's predictions for each group (central line: median; box: inter-quartile range; whiskers: minimum and maximum values). The median predictions of DeepCocrystal were significantly different between true and false classifications (*i.e.*, TP *vs.* FP, and TN *vs.* FN; Kruskal-Wallis H-test, $p < 0.05$).

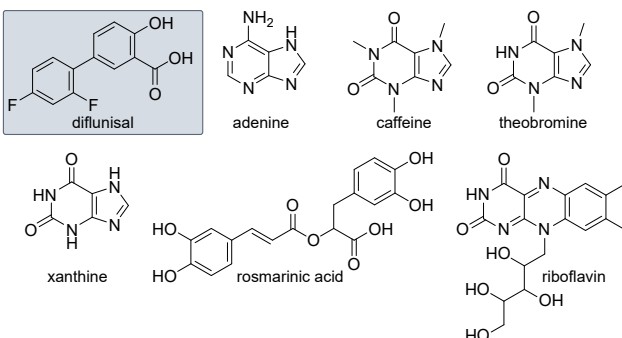

*Figure 3. Coformer candidates for diflunisal (API), selected for the prospective experimental validation.* DeepCocrystal was used to select two 'positive' predictions (adenine and caffeine), two 'negative' predictions (rosmarinic acid and riboflavin), and two high-uncertainty predictions (theobromine and xanthine) for experimental testing. The experimental tests confirmed DeepCocrystal predictions (Table 3).

agreements in the predicted class per each molecular pair is used as a measure of confidence (the higher, the better); and (b) *Standard deviation-based estimation*, whereby the standard deviation across augmented SMILES (per each pair) is computed (the lower, the better). For each approach, several thresholds of uncertainty (*i.e.*, on standard deviation or on number of agreeing predictions) were used to analyse their effect on performance, in terms of classification accuracy and number of molecules retained for prediction (Table 2).

For both uncertainty estimation strategies, DeepCocrystal performance consistently increases when using stricter thresholds (up to 10% improvement across metrics), with a progressively smaller number of predicted pairs (Table 2). Both approaches have their merits and drawbacks. Standard deviation outperforms majority voting in classification performance (up to 2% improvement), at the expanses of the number of predicted molecular pairs (57 fewer pairs). The approach to use should be chosen on a case-by-case basis, and here, we used a threshold on standard deviation equal to 0.10, to maximize prediction performance.

## 2.5. Prospective experimental application

We applied DeepCocrystal prospectively, to previously unseen molecular pairs. Diflunisal, an anti-inflammatory drug (Shen, 1983b) (Fig. 3), was selected as API, since its poor water solubility renders co-crystallization a viable strategy to enhance its bioavailability (Snetkov et al., 2021). As

*Table 3. Results of the prospective experiments guided by DeepCocrystal.* DeepCocrystal (2:7 augmentation) was used to predict the co-crystallization potential of 12 coformer candidates with the API diflunisal, and six candidates were selected for lab experiments. Mean and standard deviation of the predictions are reported (as computed on 10-fold SMILES augmentation), and a threshold on the standard deviation . The experimental outcome after lab validation is reported for the six selected molecules. Symbols indicate the outcome of the predictions and experimental validation ($\times$ = negative outcome; ? = uncertain outcome; $\checkmark$ = positive outcome).

| Tested coformer | DeepCocrystal | | Experimental Outcome |
|---|---|---|---|
| | Prediction | Outcome | |
| Adenine | $0.99 \pm 0.00$ | $\checkmark$ | $\checkmark$ |
| Caffeine | $0.99 \pm 0.01$ | $\checkmark$ | $\checkmark$ |
| Theobromine | $0.66 \pm 0.35$ | ? | $\times$ |
| Xanthine | $0.63 \pm 0.38$ | ? | $\times$ |
| Rosmarinic acid | $0.02 \pm 0.02$ | $\times$ | $\times$ |
| Riboflavin | $0.00 \pm 0.00$ | $\times$ | $\times$ |

potential coformers, we selected 12 natural products containing polyphenolic or purine moieties, due to their co-administrability and health benefits such as central nervous system stimulation, reduced risk of neurodegenerative diseases, and anti-inflammatory properties (Martínez-Pinilla et al., 2015; Yahfoufi et al., 2018; Luo et al., 2020; Rodak et al., 2021).

10-fold augmentation was performed on each SMILES strings, and the co-crystallization potential of the respective 12 API-coformer pairs was predicted with DeepCocrystal. For experimental validation, three categories of predictions were considered (Table 3): (a) top-two high-certainty, positive prediction (adenine and caffeine), (b) top-two high-certainty, negative predictions (rosmarinic acid and riboflavin), and (c) two most uncertain predictions (theobromine and xanthine). Each selected pair was tested in the lab via well-established protocols, *i.e.*, via grinding, liquid-assisted grinding, and slurry methods (Guo et al., 2021). The co-crystalization outcome was determined on the obtained powder samples, via infrared spectroscopy and solid-state nuclear magnetic resonance (*see* Materials and Methods).

All four high-certainty predictions of DeepCocrystal (adenine and caffeine as 'positive' predictions, and rosmarinic acid and riboflavin as 'negative' predictions) were confirmed experimentally (Table 3). To the best of our knowledge, the use of adenine and caffeine as coformers for diflunisal has not been previously reported. Future dissolution studies and activity assays will be needed to investigate whether this co-crystal leads to improvement in the solubility and pharmacokinetic profile of diflunisal, as observed in other caffeine-based systems (Bordignon et al., 2017; Kumar et al., 2013; Goud et al., 2012). Furthermore, both selected high-uncertainty pairs (theobromine and xanthine) did not form co-crystals (Table 3), indicating the usefulness of our un-certainty estimation approach to rule out false predictions. This experimental validation confirms the potential of Deep-Cocrystal to accelerate the discovery of novel co-crystal pairs, even with the structurally-similar selection of potential coformers selected in this study.

SMILES augmentation seemed pivotal to achieve these results. DeepCocrystal trained on canonical SMILES, in fact, predicted all purine derivate coformers as 'positive' for co-crystallization with high scores. These findings indicated that chemical language processing and SMILES augmentation allowed DeepCocrystal to capture small structural changes that might be relevant for co-crystallization. Deep-Cocrystal's capacity to correctly recognize both negative and positive pairs with high certainty underscores its potential to reduce experimental efforts in co-crystal screening and discovery.

## 3. Conclusions

Optimizing the pharmacokinetic properties of active compounds is an ever-lasting challenge in drug discovery, and co-crystallization is an attractive strategy to address this issue. However, identifying suitable co-crystallization partners for active compounds is both resource- and time-intensive. To accelerate this process, we developed DeepCocrystal, a deep chemical language processing approach designed to predict the co-crystallization of any selected molecular pairs.

This study shows the potential of DeepCocrystal to advance the state-of-the-art. DeepCocrystal owes its performance to the intriguing properties of the SMILES language, which allowed mitigating data imbalance and estimating uncertainty. By learning (and then combining) single-molecule information, DeepCocrystal learns elements of the "supramolecular language" (Lehn, 1988; Brock & Dunitz, 1994; Cragg & Cragg, 2010) of co-crystal formation. The experimental validation of DeepCocrystal further corroborated its potential and identified adenine and caffeine as two previously unreported coformers of diflunisal. These results, taken together, underscore the potential of DeepCocrystal to accelerate the discovery of co-crystallization partners.

This first-in-time adoption of the "supramolecular language" perspective with SMILES strings shows its potential for co-crystallization prediction. While this study only focused on 'two-word sentences' (*i.e.*, molecule *pairs*), our approach could be extended to supramolecular interactions among multiple molecular partners. Moreover, extensive datasets with thorough annotations on stereochemistry might further expand the co-crystal prediction ability of approaches based on SMILES strings. Ultimately, extensions of DeepCocrystal might open unexplored opportunities in supramolecular chemistry, *e.g.* for drug development(Kawakami et al., 2012), materials discovery,(Stupp & Palmer, 2014) and beyond.

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
