# OpenReview forum: "Deep Supramolecular Language Processing for Co-crystal Prediction"
_ICML.cc/2024/Workshop/ML4LMS — ML4LMS Poster_

### Official Review · Reviewer_JF15 · 2024-06-11
**Utilizing Advanced Chemical Language Processing to Forecast Co-crystal Formation**

**Rating:** 5
**Confidence:** 4

**Review:**

Summary: This paper introduces DeepCrystal, an advanced deep learning model aimed at transforming the co-crystallization phase in drug discovery. DeepCrystal utilizes the language of chemistry to depict molecules, allowing it to forecast co-crystallization with remarkable precision and speed. This approach incorporates SMILES augmentation to tackle imbalances in classes and improve its ability to generalize, distinguishing it as an innovative method in the domain. DeepCrystal emerges as a transformative force in co-crystal prediction, offering unmatched precision, ingenuity, and efficiency in the development of drug formulations. While it demonstrates strengths in prediction quality and originality, there are challenges such as interpretability, resource demands, uncertainty estimation, and the complexity of prospective studies that require further investigation. By addressing these challenges, DeepCrystal holds promise for significantly expediting drug discovery processes and opening avenues for future progress in the field.
---
Pros:

Quality:

1.1) Very Accurate: DeepCrystal is really good at guessing, with a pretty impressive 78% accuracy in real-life situations. This shows how well it can predict co-crystals.
1.2) Better Than Others: This model does better than other ones out there, proving it's great at predicting and reliable for developing new drugs.

Originality and Innovation:

2.1) Talking Chemistry: DeepCrystal does something new by using chemical language for predicting co-crystals. This fresh approach sets it apart from the usual methods.
2.2) SMILES Boost: It does something unique by using SMILES augmentation to fix imbalance issues and improve how well it can predict co-crystals.

-----
Cons:

1) Understanding Challenges:
1.1) Hard to Understand Models: Models like DeepCrystal are really complicated, which can make it tough for people to understand how they work.
1.2) Need Clearer Explanations: We need clearer explanations on how the model deals with chemical stuff so that people can understand it better.

2) Need Lots of Resources:
2.1) Complicated Way of Adding Information: Adding more information to the model in a certain way can use up a lot of computer power, which might be hard for people with limited resources.
2.2) Takes a Lot of Time to Train: When the model is being trained, doing certain things can take up a lot of time and power, which might make it harder to use the model on a big scale.

3) Problems with Estimating Uncertainty:
3.1) Hard to Understand How Sure the Model Is: The way the model figures out how sure it is about something might not be explained well, making it hard for people to make decisions based on what the model says.
3.2) Not Clear on How Sure the Model Should Be: It's not clear how sure the model should be before it says something is likely to happen, which can make it hard to trust what the model says.

4) Challenges in Future Studies:
4.1) Hard Experiments: Doing experiments in the future to find new things might be really complicated and need a lot of knowledge and resources.
4.2) Need to Be Careful: When we're looking for new things, we need to be careful not to think we found something when we didn't, especially if it looks like something we already know about.

5) Problems with Checking How Good the Model Is:
5.1) Not Checked Well Enough: We haven't tested the model against enough different things to know how good it really is, which might mean we can't trust it as much as we'd like.
5.2) Need to Test on Different Things: We should test the model on lots of different situations to see if it works well in all of them, which will help us trust it more.

---

### Official Review · Reviewer_p2X5 · 2024-06-12
**Deep understanding of the problem and good implementation**

**Rating:** 9
**Confidence:** 5

**Review:**

The paper is well-written and presents the methodology with exceptional clarity. This allows for easy replication of the experiments by other researchers. While the current selection of molecules is sufficient for a workshop paper, expanding the study to encompass a broader range could strengthen the overall contribution.

This paper presents a well-executed investigation that builds upon existing research.  The focus on clarity and reproducibility is commendable. Considering the workshop format, the current scope seems appropriate. However, for future publications, exploring a wider range of molecules could solidify the findings.